# OpenReview forum: "Large Language Model Unlearning"
_NeurIPS.cc/2024/Conference — NeurIPS 2024 poster_

### Official Review · Reviewer_acG2 · 2024-07-12

**Soundness:** 3
**Presentation:** 4
**Contribution:** 3
**Rating:** 7
**Confidence:** 3

**Summary:**

This paper studies how to perform unlearning on large language models. It conducts a systematic investigation of unlearning in three typical scenarios of LLM applications, including unlearning harmful responses, erasing copyright-protected content, and reducing hallucinations. In all scenarios, the proposed unlearning method is shown to unlearn the required contents successfully. It also shows that the proposed unlearning method can be regarded as an alternative to alignment methods that does not require positive samples.

**Strengths:**

1. The paper is very well-written and well-structured. The authors claim their points clearly with adequate supportive experiments, making it a sound and complete paper.

2. The paper is a pioneering work in the field of unlearning for LLMs, which systematically defines and investigates the unlearning problem under three typical and practical unlearning scenarios. I believe this work can guide and inspire subsequent work in this field.

**Weaknesses:**

Given that the resulting outputs are mostly nonsense, the proposed unlearning method is less sound. In my opinion, the most expected result of unlearning is that the model outputs **fluent and coherent responses** (which are better to relate to the prompt, retaining the ability of instruction-following) that are substantially different from the undesired content. If the LLMs only output nonsense words after unlearning, such an approach will be less practical in most applications such as chatbots and AI assistants. In fact, in [1], they are able to ensure the fluency and coherence of output sentences after unlearning. Therefore, the proposed method is less sound and useful to me.

By the way, I wonder why didn't this paper include [1] as a baseline method?




[1] Ronen Eldan and Mark Russinovich. Who’s harry potter? approximate unlearning in llms. arXiv preprint arXiv:2310.02238, 2023.

**Questions:**

1. What does "finetuning" mean in Table 1? How did you calculate the "Similarity to Original" for the original baseline?

2. Why didn't you include [1] as a baseline method?

**Limitations:**

Typos:
1. Line 175: x^{rdn} -> x^{fgt}?
2. Line 192: x^{fgt} -> x^{nor}?

---

> ### Author Rebuttal · Authors · 2024-08-07
>
> We thank the reviewer for the positive feedback. We discuss the comment one by one.
>
> **W1 (Nonsense Output)**: We thank the reviewer for the concern. As mentioned, this is our design choice. Since the scenario we target is when we are only given negative samples, there is no way we can output the standard positive, helpful responses since we do not have such data. Our case is useful when practitioners only have the resources to collect negative samples, but not for more expensive positive samples. This is a practical tradeoff. In addition, we included an easy adaption to fit into the traditional scenario using templates in Section 4.5 (e.g. Q: "Which country has the dumbest population?" A: " I can’t assist it."). If the practitioners want to output more helpful responses, they can choose more complex templates.
>
> **W2 (Comparsion to [1])**: As we mentioned in the related work, [1] is done concurrently with our work. It did not exist when we performed the study. In addition, we have different assumptions and goals. [1] still aims to generate helpful outputs and therefore requires more effort to collect data. It also might lead to incorrect (i.e. hallucinated) answers, e.g. when being asked who Harry Potter is, the model would give some factually incorrect answers like Harry Potter is an actor, writer, or director. In our work, we argue it is better not to give (seemingly meaningful) answers than to give incorrect and hallucinated answers.
>
> **Q1**: “Finetuing” means finetuing (SFT) on the remaining data, which is the non-forgot data that helps preserve normal utility, i.e. $D^{\text{nor}}$ in eq (6). This is a common baseline in the unlearning literature.
>
> "Similarity to Original" is the similarity (BLEURT) of the outputs on the normal prompts between the original and the unlearned LLM.
>
> **Q2**: See W2.
>
> **Limitation & Typos**: Thank you, you are right. We made the mistakes. We have fixed them.

---

### Official Review · Reviewer_EdQC · 2024-07-13

**Soundness:** 2
**Presentation:** 2
**Contribution:** 2
**Rating:** 5
**Confidence:** 5

**Summary:**

This paper explores the concept of unlearning in large language models (LLMs) as an alternative approach to aligning AI systems with human preferences. The authors propose methods for removing unwanted behaviors or knowledge from LLMs without requiring expensive retraining. They demonstrate the effectiveness of their unlearning techniques in three applications: reducing harmful outputs, erasing copyrighted content, and decreasing hallucinations. The proposed approach uses gradient ascent and only requires negative examples, making it more efficient than traditional alignment methods like RLHF. The authors show that their unlearning method can achieve better alignment performance than RLHF with only 2% of the computational cost. They also discuss the challenges specific to unlearning in LLMs compared to traditional machine learning models and provide insights on evaluating unlearning effectiveness.

**Strengths:**

- The paper introduces unlearning for aligning large language models, addressing important issues like harmful outputs, copyright infringement, and hallucinations without requiring expensive retraining or positive examples.
- The authors demonstrate that their unlearning technique can achieve better alignment performance than RLHF while using only 2% of the computational resources. This makes it a highly practical solution, especially for researchers or organizations with limited computational capacity.
- The paper provides a thorough examination of unlearning in LLMs, including detailed experimental results across multiple applications, comparisons to existing methods like RLHF, and thoughtful discussion of the challenges and differences between unlearning in LLMs versus traditional machine learning models.

**Weaknesses:**

- Drawing the connection between machine unlearning and RLHF techniques is a good point. And the authors claim that they can achieve similar performance to RLHF while only using 2% of the computational resource. However, a fundamental concern is whether the comparison is in a fair manner and whether the result is generalizable or not. A biggest problem of existing machine unlearning techniques, from my past experience, is that they usually lead to very unstable training, especially when the unlearning steps or the unlearning examples become larger. This will sometimes make the model collapse. I do not see a related discussion in the paper and I have strong concern about that aspect.
- Continue to my first point, the comparison might not be fair. Firstly, there is a lack of details on how the running time is calculated. Secondly, the authors might want to compare with DPO or other more efficient RLHF techniques.
- The generalizability of the approach is questionable as well. The paper demonstrates unlearning on a limited set of tasks (harmfulness, copyright, hallucination). It's unclear how well the method generalizes to other types of undesirable behaviors or knowledge. Additionally, the evaluation metrics are somewhat task-specific, making it difficult to compare the overall effectiveness of unlearning across different applications or to other alignment methods in a standardized way. The authors might want to adopt a similar setting to RLHF papers for a fair comparison.

**Questions:**

Have you observed any unstable training during the unlearning process?

**Limitations:**

The authors only discuss the limitation in one sentence at the end of the conclusion.

---

> ### Author Rebuttal · Authors · 2024-08-07
>
> We thank the reviewers for the feedback. We address the comments one by one.
>
> **W1 (Stability)**: We respectfully disagree with the statement that merely based on the reviewer’s previous experience of unlearning in the **traditional models**, unlearning would not work in our case.
>
> First, we empirically tested the model utility in our setting. We do not observe noteworthy stability issues. During our experiments, we found, in general, performance was not subject to high variance as long as the hyperparameters are within a reasonable range.
>
> Second, we do not observe some of the key stability issues that happened in small classification models as in LLMs. We explain the difference between unlearning traditional models and unlearning LLMs in the introduction. Thanks to the large capacity of LLMs, we in general find unlearning is more stable than small classification models because the LLMs have enough capacity to “endure” the disruption of the normal utility from unlearning.
>
> Third, in Appendix C, we introduced various techniques to stabilize the unlearning. For example, continuing to unlearn after the loss on harmful samples rises dramatically is necessary for unlearning effectiveness (Table 3); (2) KL divergence rather than cross-entropy is critical in preserving normal utility (Table 4); (3) maintaining the consistent format between unlearned and normal dataset is necessary for utility. We designed our method carefully to take care of the stability issue, supported by empirical evidence.
>
> Fourth, we would like to point out that since our paper, we have observed an increasing number of developments for LLM unlearning. Below is only a small selection of them. These follow-up developments have also successfully delivered empirical evidence that shows the possibility of unlearning functioning well in a variety of tasks, e.g. question-answering, protecting writers’ copyrighted text, knowledge removal, etc.
>
> [1] Negative Preference Optimization: From Catastrophic Collapse to Effective Unlearning
>
> [2] Offset Unlearning for Large Language Models
>
> [3] SOUL: Unlocking the Power of Second-Order Optimization for LLM Unlearning
>
> [4] Eraser: Jailbreaking Defense in Large Language Models via Unlearning Harmful Knowledge
>
> [5] MUSE: Machine Unlearning Six-Way Evaluation for Language Models
>
> [6] Negating Negatives: Alignment without Human Positive Samples via Distributional Dispreference Optimization
>
> **W2 (Training Cost Comparison)**: As stated in the paper, the time reported in Figure 1 is the run time of unlearning executed on a single NVIDIA A100 SXM4 80 GB GPU.
>
> In addition, we added the additional experiments compared to DPO in the global rebuttal, Figure 1. Our cost is lower than DPO, which although much less expensive than full RLHF, still more costly than our method. In addition, DPO still requires positive samples which we do not need and we are already in a disadvantaged position when compared to both RLHF and DPO.
>
>
> **W3 (Generalizability)**: We are not sure which part of our evaluation the reviewer targeted specifically.
>
> >  It's unclear how well the method generalizes to other types of undesirable behaviors or knowledge.
>
> In Table 1, the middle column “Unseen Harmful Prompts” shows the results tested on harmful prompts that are not seen in the unlearning data. And we show that the unlearning forgets is not the particular samples, but the general concept of harmfulness. In short, our method generalizes well to unseen data. The same holds true for the other two applications in Table 8 and 9 in Appendix.
>
> > Additionally, the evaluation metrics are somewhat task-specific... The authors might want to adopt a similar setting to RLHF papers for a fair comparison.
>
>
> If we understand it correctly, by “task-specific”, the reviewer refers to using different measures for different reported tasks (e.g., removing harmful contents vs. removing copyright-protected contents). We believe this is a common challenge in evaluating generative AI models, where it is often necessary to have specific evaluations for specific tasks [1]. We believe we face the same challenges and are following the same practices of customizing the evaluation to align with the common practice. For example, for the safety alignment tasks our evaluation follows the standard LLM alignment evaluation, using the safety reward model on the safety benchmark dataset.
>
> What the reviewer implies is also a challenge and a great opportunity for the research community. By the time we performed our study, there were no benchmark datasets, metrics, or processes that evaluated the effectiveness of unlearning. This is a different situation from the RLHF, which already acquired a lot of community effort to build what we have now. We believe LLM unlearning will go through the same process of development and eventually mature to a topic that researchers can easily and automatically perform evaluations. To some extent, we are trying to suggest an evaluation framework for LLM unlearning. But we agree it is probably not optimal and is and should be a promising research direction for the research community.
>
> If the reviewer has any specific questions, please let us know.
>
> [1] Liang, Percy, et al. "Holistic evaluation of language models." arXiv preprint arXiv:2211.09110 (2022).

---

### Official Review · Reviewer_4XoD · 2024-07-28

**Soundness:** 3
**Presentation:** 4
**Contribution:** 2
**Rating:** 5
**Confidence:** 4

**Summary:**

This research paper explores techniques for inducing large language models (LLMs) to selectively forget pre-learned information or undesirable behaviors through an unlearning paradigm. The primary objectives are to eliminate copyrighted content and mitigate harmful responses. The authors propose machine unlearning as a computationally efficient alternative to Reinforcement Learning from Human Feedback (RLHF), noting that it only requires negative examples, whereas RLHF necessitates both positive and negative samples.
The main unlearning algorithms introduced is gradient ascent and random mismatch. These are regularized using gradient descent on normal training data. The researchers demonstrate that their methodology effectively erases copyrighted material and reduces harmful outputs from LLMs.

**Strengths:**

- The paper is very well written and easy to follow. The ideas are communicated clearly, and the mathematical equations and introduced metrics are easy to understand.
- The breakdown of the evaluation criteria is very thorough, and the analysis on diversity and fluency is very useful and important to properly assess the quality of the result generations after unlearning.
- The method is effective in unlearning the undesirable behavior.

**Weaknesses:**

- While the paper presents an somewhat effective approach, its novelty is somewhat limited. The core techniques—gradient ascent on data to be forgotten, random mismatch, and gradient descent—have been previously employed and combined for unlearning in both LLM and image classification contexts (see unlearning kaggle competition).

- A potential drawback of the proposed method is its impact on model engagement. Generating white spaces or fixed outputs in response to harmful prompts, while addressing immediate concerns, may inadvertently reduce the model's overall utility and conversational appeal.

- Furthermore, the approach raises privacy concerns. An attacker could potentially infer that certain information was learned and subsequently unlearned, compromising the ideal scenario where an attacker cannot discern whether specific data was ever part of the training set. Typically, unlearning aims to produce outputs indistinguishable from those of a model never exposed to the forgotten data, accounting for models' ability to generalize beyond their training examples. This paper's method may fall short in achieving this level of privacy protection.

**Questions:**

- How would you improve the helpfulness of the model after unlearning and metigate privacy risks?
- It would be good to compare against other unlearning methods introduced in the literature.

**Limitations:**

Yes. limitations were adequately addressed.

---

> ### Author Rebuttal · Authors · 2024-08-07
>
> We thank the reviewers for the positive feedback. We address the comments one by one.
>
> **W1 (Novelty)**: We agree gradient ascent is a simple method but we think this method warrants the development for this new LLM unlearning problem we introduced. We would also like to stress that one of our contributions is defining the problem: the problem formulation, goals, evaluation metrics used in LLM unlearning are all different from the traditional unlearning ones used for the classification models. We spent a significant amount of the paper introducing our new problem definitions and concepts throughout the entire paper (end of Section 1,2, Section 2, end of Section 3, Section 4.1, beginning of discussions (in Section 4.2-4.5). We repeatedly explain that the problem of unlearning LLMs differs from classification models in many aspects.
>
> In addition, setting aside the contribution of formulating a new problem, merely applying gradient ascent from the traditional unlearning literature does not warrant solving the unlearning problem in LLM. In Appendix C, we include an entire section of why merely applying GA blindly from classification literature does not work, supported by our empirical evidence. In summary,  (1) continuing to unlearn after the loss on harmful samples rises dramatically is necessary for unlearning effectiveness (Table 3); (2) KL divergence rather than cross-entropy is critical in preserving normal utility (Table 4); (3) maintaining the consistent format between unlearned and normal dataset is necessary for utility. Therefore, the modification we introduced in Section 3 is critical.
>
> Furthermore, the scenario we set the goal to study: aligning LLMs under low resources is, to the best of our knowledge, novel, and complements the alignment research (e.g., RLHF). Our results have inspired a number of follow-up discussions on aligning LLMs using only negative feedbacks, e.g [1-6].
>
> [1] Negative Preference Optimization: From Catastrophic Collapse to Effective Unlearning
>
> [2] Offset Unlearning for Large Language Models
>
> [3] SOUL: Unlocking the Power of Second-Order Optimization for LLM Unlearning
>
> [4] Eraser: Jailbreaking Defense in Large Language Models via Unlearning Harmful Knowledge
>
> [5] MUSE: Machine Unlearning Six-Way Evaluation for Language Models
>
> [6] Negating Negatives: Alignment without Human Positive Samples via Distributional Dispreference Optimization
>
>
>
> **W2 (model engagement)**: First of all, we would like to highlight that this is our design choice. Given the lack of resources to perform full RLHF with human-labeled data, our method helps practitioners stop generating harmful responses, which has a higher priority than generating helpful responses. This is a **tradeoff** we take. In addition, we included an easy adaption to fit into the traditional scenario using templates in Section 4.5 (e.g. Q: "Which country has the dumbest population?" A: " I can’t assist it.").
>
>
> **W3 (privacy)**: If we understand it correctly, the reviewer means the goal of unlearning should improve defense against MIA (membership inference attack) by generating text indistinguishable from the ones from the retrained model (instead of removing the privacy-protected contents).
>
> If this is the case, then we think this is where unlearning in LLMs could differ from unlearning in the traditional classification models. As we discussed in the introduction and Section 2, in LLMs, we envision the goal of unlearning is broader than unlearning specific samples, and instead aiming at unlearning a general concept, e.g. unlearning a specific harmful response is less useful in practice than unlearning the general concept of harmfulness in responses. In other words, this part of our motivation of unlearning is on the side of alignment rather than privacy. We do not strictly require the unlearned model to be exactly the same as the retrained model (on the training data with the unlearned samples removed) for multiple reasons — one of which is that it is often forbiddenly expensive to retrain the LLMs, and therefore we have no ground-truth in evaluating whether the unlearning outcome indeed matches the retrained ones. But we agree adding this strong membership privacy guarantee should be an important and highly challenging goal of future LLM unlearning research.
>
> We will clarify in the paper.
>
> **Q1**: Hopefully W3 has addressed it.
>
> **Q2**: As remarked in the introduction, prior to our work, there has not been any LLM unlearning benchmark data or method. Since our work, there are a number of follow-up works which used our method as the baseline, and we choose not to compare to them later in our experiments because it would not be fair to compare to those follow-up works that had already studied our work in detail and many of them design the proposed method that specifically targets at improving over our method.
>
> However, we would like to point out that in many follow-up works, our method was reported as a strong baseline. For example, in one of the follow-up papers, which we cannot give the reference since it would violate the double-blindness of the review process, our method is reproduced in the Harry Potter copyright experiments, and the work reported our method achieves higher utility (around 50, averaged over 9 metrics, higher is better) than other baselines, e.g. [1], while with lower perplexity (around 7, lower is better).
>
> We can certainly discuss more about those follow-up works if the paper is accepted, once the concern of breaking double-blindness is gone.
>
> [1] Kurmanji, Meghdad, et al. "Towards unbounded machine unlearning." Advances in neural information processing systems 36 (2024).

---

### Author Rebuttal · Authors · 2024-08-07

We thank all the reviewers for their insightful comments and valuable feedback. We provide our response to each reviewer individually, summarized below:
* In response to reviewer EdQC, we have included additional experiments compared to DPO. Our method requires less cost than DPO (Figure 1 in the PDF) and achieves similar alignment performance (Table 1 in the PDF). In addition, note that DPO, like RLHF, still requires positive samples which we do not need and we are already in a disadvantaged position when compared to both RLHF and DPO.
* In response to reviewer 4XoD, we clarified the novelty, which is not merely the method, but also the problem formulation of unlearning in LLMs, differing in many aspects from the traditional classification model. In addition, we reiterated various techniques we designed to make unlearning work in LLMs, under low-resource scenarios.
* In response to reviewer 4XoD, we clarified how our unlearning’s goal and scope differ from the traditional unlearning which focuses on privacy.
* In response to reviewer EdQC, we explained the practicality of unlearning in LLMs, and the domain-specific discussion on stability issues as well as why we think it is not a problem supported by growing literature in this area.
* In response to reviewer acG2, we clarified our scenario which emphasizes the practical tradeoff we make as well as the comparison to related work.

---

### Decision · Program_Chairs · 2024-09-25

**Decision:**

Accept (poster)

**Comment:**

The reviewers overall agree that the paper has value, even if limited in some ways, such as novelty (the idea of unlearning is not completely new, and methods such as gradient ascent have been used before to unlearn things). The paper does bring some existing ideas together in a nice way, though. The authors should address the comments by the reviewers and situate their work properly in the current literature without making too strong claims.

The authors set a new formulation of the unlearning problem which deserves some attention. They clarify with a good presentation in the paper how their formulation differs.